# Through-Ice Acoustic Source Tracking Using Vision Transformers with Ordinal Classification

**DOI:** 10.3390/s22134703

**Published:** 2022-06-22

**Authors:** Steven Whitaker, Andrew Barnard, George D. Anderson, Timothy C. Havens

**Affiliations:** 1Department of Electrical and Computer Engineering, Michigan Technological University, Houghton, MI 49931, USA; sjwhitak@mtu.edu; 2Graduate Program in Acoustics, The Pennsylvania State University, University Park, PA 16802, USA; barnard@psu.edu; 3Naval Undersea Warfare Center, Newport, RI 02841, USA; george.d.anderson@navy.mil; 4Department of Computer Science, Michigan Technological University, Houghton, MI 49931, USA

**Keywords:** ice acoustics, localization, ordinal classification, Vision Transformers

## Abstract

Ice environments pose challenges for conventional underwater acoustic localization techniques due to their multipath and non-linear nature. In this paper, we compare different deep learning networks, such as Transformers, Convolutional Neural Networks (CNNs), Long Short-Term Memory (LSTM) networks, and Vision Transformers (ViTs), for passive localization and tracking of single moving, on-ice acoustic sources using two underwater acoustic vector sensors. We incorporate ordinal classification as a localization approach and compare the results with other standard methods. We conduct experiments passively recording the acoustic signature of an anthropogenic source on the ice and analyze these data. The results demonstrate that Vision Transformers are a strong contender for tracking moving acoustic sources on ice. Additionally, we show that classification as a localization technique can outperform regression for networks more suited for classification, such as the CNN and ViTs.

## 1. Introduction

Acoustic source localization is important in underwater acoustics. In underwater environments, acoustic frequencies propagate long distances, which permits acoustic analysis to be ideal for localization. Localizing a source is beneficial in numerous applications: search and rescue for the coast guard, tracking ships for merchant shipping, and situational awareness for military purposes, to name a few. In a deep water environment, such as the ocean, varying sound speed profiles present challenges in properly simulating the environment [1,2,3]. In ice environments, even more challenges arise: multi-path, scattering fields, interference patterns with a reflective ice surface, non-linear propagation through the ice, and a temporally changing field [4,5]. Additionally, shallow-depth, narrow, ice-covered waveguide environments (e.g., a frozen river or a canal) generate more multipath reflections on the bottom and edges of the environment. These narrow ice environments are important for tracking snowmobiles or other anthropogenic sources on or under the ice. Therefore, Machine Learning (ML) is a promising method to investigate for such a highly complicated environment that can incorporate all the complex water environment and the complex ice environment.

ML has been used previously in acoustic localization approaches with great results [3,6,7,8,9,10]. Long Short-Term Memory (LSTM) neural networks have been shown to analyze time-series acoustic data with success [9,10,11,12]. LSTM is designed to analyze data with time dependence [11], but its computational complexity causes difficulty in training large networks, which is shown in Section 3. A newer concept is to utilize Vision Transformer (ViT) architectures [13,14]. The ViT is a modified version of a Transformer neural network [15,16] where the ViT is specialized for data with a large number of dimensions, e.g., acoustic spectrum data. The ViT has been used extensively in computer vision and image analysis [13,17,18], but to date, there has been no paper published on ViT-based localization for through-ice or underwater acoustic localization.

To combine multiple state-of-the-art concepts, our previous work showed that localization framed as a classification problem outperforms regression [10]. With a constrained area of interest, the regression values can be transformed to be classes that represent a grid of positions, and then, the neural network estimates these classes. This classification is an alternative to localizing the source with regression. With respect to our prior research [10], we tested the claims proposed in the classification method with new data and show that the proposed classification method has more nuanced results. We show that networks suited for classification problems show better localization performance with the proposed method, while networks suited for regression problems better localize the source with a regression loss. We validated this claim with newly conducted experiments on ice, a larger training dataset, and new, state-of-the-art neural network architectures.

We show the results of these algorithms with newly recorded data for localizing and tracking on-ice snowmobiles on the Keweenaw Waterway in Houghton, Michigan, by comparing the four described neural network architectures—Convolutional Neural Networks (CNNs), LSTMs, Transformers, and ViTs—with three loss functions: regression, categorical classification, and ordinal classification. We first provide an understanding for how our data are recorded to explain which properties our ML algorithms will exploit.

## 2. Materials and Methods

To record our acoustic source, we used an Acoustic Vector Sensor (AVS), which is capable of recording acoustic pressure and acoustic particle velocity (or acceleration) within a single sensor module [19]. Our experiments used two Meggitt *VS–209* underwater pressure and particle acceleration (*pa*-type) AVSs [20], which record acoustic pressure and acoustic acceleration simultaneously. A *pa*-type AVS consists of a hydrophone and a triaxial accelerometer in the same module and is a good choice for the experiments in this paper because the accelerometers’ bandwidth reaches higher frequencies than a *pu*-type (pressure and particle velocity) AVS [20]. A snowmobile’s response is a relatively broadband signal; hence, we can record more of the signal source’s frequency domain signature. The Meggitt *VS–209* has a bandwidth up to 8000 Hz, and the snowmobile’s broadband signal goes up to 10,000 Hz [21,22], which is also seen in the raw data in Section 2.7.1.

### 2.1. Acoustic Post-Processing

A single *pa*-type AVSs generates four time-series data streams. Using a single sensor, we can produce an angle measurement by post-processing these time-series streams. This angle measurement, the Direction Of Arrival (DOA), tells us from which direction the sound arrives, no matter if the sound is from the acoustic source we are trying to track or if the sound is from other sources, e.g., waves crashing, biometrics, or anthropogenic sources that are not our target, to name a few. Each AVS produces its own acoustic intensity, *I*, with post-processing [19]:(1)$$I_{x,y,z}\left(f,t\right)=\frac{P\left(f,t\right)A_{x,y,z}^{*}\left(f,t\right)}{j2\pi f},$$
where P(f,t) is the acoustic pressure in the frequency domain at time *t* (i.e., *P* is the Short-Time Fourier Transform (STFT) of the pressure time-series p(t)), $A_{x,y,z}\left(f,t\right)$ is the three-dimensional acoustic accelerations in the three axial directions, x,y,z, from the AVS accelerometer in the frequency domain at time *t*, *f* is frequency, ${}^{*}$ is the complex conjugate, and $j=\sqrt{-1}$. The *VS-209* contains a coordinate transform to transform the $I_{x,y,z}$ positions into a “global” coordinate system that is aligned from Earth’s magnetic field and Earth’s gravitational field:(2)$$I_{\mathrm{west},\mathrm{north},\mathrm{up}}\left(f,t\right)=Q^{T}I_{x,y,z}\left(f,t\right),$$
where $Q^{T}$ is the coordinate transform defined in the *VS-209* system manual. Acoustic intensity is then used for azimuth calculation via
(3)$$\theta \left(f,t\right)=\arctan\frac{I_{west}\left(f,t\right)}{I_{north}\left(f,t\right)},$$
where θ is the azimuth DOA of the acoustic source; east is 0 degrees, and north is 90 degrees. When using an STFT, θ is a spectrum of angles, called an *azigram* [23]. From this point on, we will consider azigrams as a two-dimensional image, where $\theta_{f,t}=\theta \left(f,t\right)$, which matches well with the computer vision background of deep networks. The vector $\theta_{t}$ denotes the column of the matrix θ at time *t*.

Thus far, our post-processing has yet to deal with any aspect of multi-path, scattering fields, interference patterns, or reflections prevalent in this signal, i.e., interferences are still incorporated in θ. Suppose a target were not generating a signal at some time, e.g., the target has moved out of range of the sensors or the target powered off its noise source. In this scenario, angle measurements would come from the ambient background, which often presents a localized noise or “noise coming from certain angles.” Because θ is a noisy signal, we need to further process this signal. We will use ML to handle the noise, which is an excellent algorithm for working with high-dimensional and noisy data. Specifically, we discuss four different neural network approaches. The four neural networks we investigated are: (i) a CNN, (ii) an LSTM neural network, (iii) a Transformer neural network [15], and (iv) a ViT [13]. Let us now describe each of these networks in detail and adapt these different networks to our localization problem.

### 2.2. Convolutional Neural Network

The CNN performs convolution operations on the input signal, and in this regard, we perform a 2-dimensional convolution along both the frequency and time:(4)Y=W★θ,
where ★ is the convolution operation, *W* are the trainable parameters in the CNN, and *Y* is the output of a single CNN operation. The convolution operation, W★θ:(5)$$Y_{f,t}=\sum_{i=0}^{F}\sum_{j=0}^{T}W_{i,j}\theta_{f-i,t-j},$$
elucidates local relations spanning across the time domain, t∈[0,T], and the frequency domain, f∈[0,F]. The kernel size—i.e., the dimensionality of *W*—is a parameter that can be adjusted to allow larger relations across time and frequency. With an activation function, such as tanh or ReLU: (6)$$\mathrm{ReLU}\left(x\right)=\{\begin{array}{ll} x & \mathrm{if}\;x\geq 0\\ 0 & \mathrm{otherwise}, \end{array}$$
surrounding Equation (Equation 4), the CNN is now a non-linear transform. CNN layers are extremely powerful in a Deep Neural Network (DNN) [24], but there are some pitfalls. The CNN handles spatially localized features, but the CNN lacks any temporal aspect, i.e., any long-term or temporal relations are not represented or handled. With a CNN, each input is independent of the next. Our data are not independent of each other, since our data are time-series and the position of an acoustic source traveling by the sensors is dependent on its previous position; that is, real-world sources have temporal correlation in their acoustic signal. We incorporated this temporal information with an LSTM.

### 2.3. Long Short-Term Memory Neural Network

LSTMs address some of the weaknesses of CNNs for time-series data. They look into the temporal and long-term relations with the short-term hidden state, $h_{t-1}$, and long-term candidate state, $c_{t-1}$, in each LSTM cell, seen in Figure 1 [11].

The equations derived from Figure 1 consist of “gating” the logical flow. For example, the “forget” gate, $f_{t}$, limits how much the long-term candidate state, $c_{t-1}$, is incorporated into the output, $h_{t}$. The other two gates operate similarly; the “input” gate, $i_{t}$, limits the effect of input data, $h_{t-1}$ and $x_{t}$, and the “output” gate; $o_{t}$, limits the effect of total data on the output, $h_{t}$. This is reminiscent of a Kalman filter’s capability to adjust the estimate based on its prior knowledge; however, an LSTM can also adjust the output of its prior knowledge in addition to the new measurements. The equations for these gates are
(7)$$\begin{array}{cc} i_{t} & =\sigma (W_{i}{\left[h_{t-1}^{T}\;x_{t}^{T}\right]}^{T}+b_{i}) \end{array}$$
(8)$$\begin{array}{cc} f_{t} & =\sigma (W_{f}{\left[h_{t-1}^{T}\;x_{t}^{T}\right]}^{T}+b_{f}) \end{array}$$
(9)$$\begin{array}{cc} o_{t} & =\sigma (W_{o}{\left[h_{t-1}^{T}\;x_{t}^{T}\right]}^{T}+b_{o}), \end{array}$$
where matrices *W* and vectors b correspond to the trainable gate parameters (input, forget, output), σ (shown in Figure 1) is the sigmoid activation function, $1/(1+e^{-x})$, and $\left[\cdot \right]$ is a concatenation of the vectors. The equations for the LSTM outputs are
(10)$$\begin{array}{cc} c_{t} & =f_{t}\circ c_{t-1}+i_{t}\circ \tanh\left({\left[h_{t-1}^{T}\;x_{t}^{T}\right]}^{T}\right) \end{array}$$
(11)$$\begin{array}{cc} h_{t} & =o_{t}\circ \tanh\left(c_{t}\right), \end{array}$$
where ∘ is an elementwise multiplication.

LSTMs are “chained together” successively using the LSTM cells in Figure 1; that is, the output, Equation (Equation 10), of the previous LSTM cell is the input to the next LSTM cell. This chaining can be used for long-term memory in the system. The vectors, c and h, are stateful values of the LSTM, i.e., they are dependent on the input data to and internal weights of the LSTM cell (and subsequently, all previous LSTM cells). The LSTM is dependent on its previous state because the outputs of the previous LSTM cell is the input of the next LSTM cell (along with $x_{t}$), and so, the mathematical operations are sequential for each LSTM cell. This means the LSTM operations cannot be computed in parallel. Because of this limitation, LSTMs inherently train slower because other neural network architectures can utilize GPU parallel processing more. The training speeds are shown in Section 3. The Transformer architecture attempts to avoid the LSTM’s sequential computational processing while keeping temporal relations with attention-based networks, which we explain now.

### 2.4. Transformers

Since Transformers have seen promising results in natural language processing [15] and image classification [13], we anticipate Transformers and Transformer variants will perform well in spectrum analysis. Transformers utilize self-attention [25], where self-attention is defined as the normalized dot product:(12)$$\mathrm{attention}\left(\theta \right)=\mathrm{softmax}\left(\frac{QK^{T}}{\sqrt{d}}\right)V,$$
where *Q*, *K*, and *V* are the projected query, key, and value tensors: $Q=W_{Q}\theta$, $K=W_{K}\theta$, $V=W_{V}\theta$, where *W* are trainable parameters [15]. θ are the input data, i.e., the azigram image. The scaling parameter, $\sqrt{d}$, is found to better normalize the data, suggested in [15]. For our data, d=512, the number of frequency bins in the azigram. The softmax function:(13)$$\mathrm{softmax}\left(x\right)=\frac{e^{x}}{\sum_{k=1}^{K}e^{x_{k}}},$$
normalizes the data such that ∑softmax(x)=1. Multi-Head Attention (MHA) calculates Equation (Equation 12) multiple times to permit different attention interconnections with the same data. MHA allows for multiple relations to be found within the same layer in the Transformer.

The Transformer then projects the results of Equation (Equation 12) by
(14)y=ϕ(θ+attention(θ))+θ+attention(θ),
where ϕ is a projection operator; in our case, ϕ is a fully connected neural network. Figure 2 illustrates Equation (Equation 14), along with the additional normalizing used within the Transformer architecture. The normalization ensures invariance to scale differences in the feature space, as suggested in [15].

The benefit to self-attention is any abstract relation can be represented within a sample along the temporal and frequency dimensions of our azigram data [15]. This abstraction results in a more broadly applicable CNN. Additionally, a Transformer outperforms the LSTM in training speed with its capability to train in parallel, rather than sequentially, since all the operations in Equation (Equation 12) are independent of one another. With a spectrum, the Transformer finds attention across all possible azimuth, θ, values, which generates a massive matrix of attended values. If there are a large number of dimensions to which the Transformer attends, there is a large scope to search. The vanilla Transformer struggles to analyze such high-dimensional data. The Vision Transformer better handles this issue using positional embedding.

### 2.5. Vision Transformers

A ViT is a modified Transformer that encodes a highly dimensional image (in our case, an azigram) into smaller patches within its position embedded into the Transformer. A positional embedding is added; Figure 3 shows a setup where the spectrum data are chunked into the Transformer with the positions embedded [13].

For example, with 16 positional embeddings and a 512×512 image, the ViT can embed 16 images of size 128×128 in a 4×4 grid pattern, enclosing the 512×512 azimuth input. The positional embedding is a trainable parameter, so this example is not used in the network itself, but rather as a simple representation of the positional embeddings being adjusted by the ViT. Generalizing this example, we change from $N^{2}$ parameters with the Transformer to $N^{2}/M$ attention values with the ViT when each of *M* embeddings are the same size [13,17]. The reduced attention relations are beneficial for data with large numbers of dimensions, the benefits of which are shown in Section 3.

### 2.6. Loss Functions

With each of the networks described, we now turn to defining our separate loss functions for localizing our target, the first of which is the “standard,” or most common loss function for localization: regression.

#### 2.6.1. Regression

A regression loss function is typically an $l^{p}$-norm equation, commonly the *Mean-Squared Error* (MSE) or *Root-Mean-Squared Error* (RMSE). For example, the RMSE is
(15)$$L={∥p^{*}-p^{true}∥}_{2},$$
where $p^{*}$ and $p^{true}$ are the predicted target position and true target position, respectively.

Fundamental faults of a typical regression loss function are the lack of predicted certainty of the results and the inability to constrain predictions in a nuanced manner. It is of importance in some applications to know how confident the localization is, e.g., tracking the signal while it travels out of the sensors’ effective *Signal-to-Noise Ratio* (SNR) ranges. Additionally, a more constrained field of predicted values can benefit performance if one is predicting in a predetermined area (such as the bend of a river) [10]. As such, we will now propose a classification approach whereby we predict locations on a predetermined grid and then aggregate to predict a location. This method also provides the confidence or uncertainty of the prediction.

#### 2.6.2. Categorical Classification

A regression loss function provides no measure of confidence and, thus, simply provides a localization estimate even when the network is presented with pure noise. This is not adequate for a generalized solution for localization. In contrast, categorical classification was initially investigated as a method to not only provide a location prediction, but also the confidence in this prediction [9]. Another benefit of a classification approach to localization is that the localization region can be predefined, i.e., a neural network with a classification output can be designed to only predict at specific regions (e.g., water, and not beach). Neural networks with a regression output predict *any* output, and this may not be viable in a real-world scenario, such as a water vessel being constrained to within the banks of a river.

Our categorical classification method manipulates a grid mapping of locations, then predicts the classes in a manner where one can determine the certainty of the network prediction. We use a soft label classification equation: (16)$$y_{k}=\frac{1}{\Delta }\prod_{d=1}^{D}\{\begin{array}{cc} \Delta -\left|p_{d}-{\left(c_{k}\right)}_{d}\right| & \mathrm{if}\;|y_{d}-{\left(c_{k}\right)}_{d}|\leq \Delta \\ 0 & \mathrm{otherwise}, \end{array}$$
where p is the true target position, $c_{k}$ is the vector location of the *k*th classification grid position, Δ is a distance threshold, and $y_{k}$ is the soft-labeled true target corresponding to the classification grid positions, $c_{k}$. To generate $c_{k}$, a *D*-dimensional grid of positions is created that correspond to positions in the real world.

Our data are 2D in nature with variations in only latitude and longitude; thus, D=2. To simplify calculations, the distance between adjacent classes—i.e., grid positions $c_{k}$—is normalized to be 1. To ensure that only adjacent classes in $c_{k}$ to any given ground truth location p are non-zero-valued, we chose Δ<1. For example, in Figure 4, the green circles would be the only elements of y that are non-zero-valued.

Figure 4 shows a position, p, among the 4 closest grid points, $c_{1}$, $c_{2}$, $c_{3}$, and $c_{4}$. The associated soft label $y_{k}$ for each of these grid locations is inversely proportional to the distance from class location $c_{k}$ and p, described in Equation (Equation 16). As such, the upper-right truth label $y_{2}$ of the 4 classes in Figure 4 has the smallest soft label, and the lower-left truth label $y_{3}$ has the highest value.

If a position, p, is equidistant to all surrounding classes, the non-zero values of y are all equal. Additionally, suppose the ground truth position, p, is positioned directly on a class, $c_{k}$, then
(17)$$y_{k}=\{\begin{array}{ll} 1 & p=c_{k}\\ 0 & \mathrm{otherwise}. \end{array}$$

When converting back to a continuous location space, each classification grid is defined on specific coordinates; thus, we can yield the original position,
(18)$$p=\sum_{k=1}^{N}y_{k}g_{k}$$
where g corresponds to the “real-world” grid mapping to the classification locations, y. For example, g can be a grid of GPS coordinates or a grid of pixel positions in an image.

Soft classification is also useful when the truth data are uncertain (e.g., a distribution) as opposed to classifying a single class for the truth data. For the purposes of this paper, the errors in the truth data and their distribution are not considered because the uncertainties of our truth data (within 2.5 m [26]) are smaller than the the distances between each class (28 m), i.e., there are no benefits to adding uncertainty when localizing our target.

When calculating Equation (Equation 16) for our target positions, we may find that the locations are constrained to smaller regions of the full rectangular grid; thus, the grid can be adjusted such that only certain locations are used. The dimensionality of the prediction can be reduced by removing classes—i.e., grid locations. For example, these removed locations can materialize if there are physical obstructions at those locations. Additionally, we observed that background noise often will manifest as position estimates that are outside the region of interest (i.e., the water body). In the future, we will look at how we can specifically design our algorithms to identify background noise when no source to track is present, but for this study, we simply constrained the classification grid to within the banks of the region of study (a canal), where Figure 5 shows the regions outside the banks. Because our experiments are simulating environments of a ship in the water or a snowmobile traveling across the ice, we can constrain the classification grids to regions where the acoustic sources can only reach physically. These constrains are a benefit to the classification approach to localization, but further constraints could bias our results to the data.

An example of the grid location classes for a 10×10 grid is shown in Figure 5. The “out of bounds” labels on the bottom-left corner in Figure 5 correspond to outside the banks of the Keweenaw Waterway, and no data are present on these grid location classes.

When the classification labels are represented as soft-labeled grid locations, we can use an MSE loss between each dimension,
(19)$$L=\frac{1}{K}\sum_{k=1}^{K}{\left(y_{k}-{\hat{y}}_{k}\right)}^{2},$$
where *K* corresponds to the number of classes, y is the true (soft) classification label vector, and $\hat{y}$ is the predicted (soft) classification label vector.

The weakness of classifying with grid representation and the categorical loss in Equation (Equation 19) is their ordinal (spatial) nature is not fully considered. If the network were to predict an incorrect location physically close to the true location, this should not be equally penalized to predicting a location far away from the true location. Categorical classification fails to represent this; hence, we describe how to extend this idea to ordinal classification for localization.

#### 2.6.3. Ordinal Classification

To give an example of the impetus for the ordinal (spatial) property of the classification grid, consider a prediction at position (0,1) when the correct class is at position (0,0); clearly, this incorrect prediction is not as poor as predicting at the position (99,99). Categorical loss would consider these two incorrect predictions to be equally poor, but our proposed ordinal loss properly represents the relative error of each of these predictions. Extending ordinality to the classification problem introduces complexity, as the loss function becomes more advanced, but this added complexity better represents our localization problem [27]. Our proposed ordinal loss function gives lower weight to closer predictions to the truth [9],
(20)$$L=\frac{1}{K}y^{T}W{\left(y-\hat{y}\right)}^{2},$$
where *W* is a weighting matrix and ${\left(\cdot \right)}^{2}$ indicates an elementwise square operation in this equation. The weighting matrix, *W*, $W_{i,j}={∥c_{i}-c_{j}∥}_{2}$, is a K×K matrix of the pairwise $l_{2}$-norm distances between each grid position c. One can think of the product $y^{T}W$ as the weighted mean distance of each grid location to the predicted location represented by y. This is then multiplied by the vector that represents the squared differences between the predicted location y and the truth $\hat{y}$. Consider the following example.

Consider a 2×2 grid of locations, where $c_{k}$, k=1,2,3,4, represent the grid positions [(0,0),(0,1),(1,0),(1,1)]. In this scenario, the weight matrix is
$$W=\left[\begin{array}{cccc} 0 & 1 & 1 & \sqrt{2}\\ 1 & 0 & \sqrt{2} & 1\\ 1 & \sqrt{2} & 0 & 1\\ \sqrt{2} & 1 & 1 & 0 \end{array}\right].$$

Suppose $y={\left[0,1,0,0\right]}^{T}$ (representing a prediction at position (0,1)) and $\hat{y}={\left[0,0,1,0\right]}^{T}$ (representing the ground truth position (1,0)). The product $y^{T}W=\left[1,0,\sqrt{2},1\right]$, and the loss is
$$\begin{array}{cc} L & =\frac{1}{4}\left[1,0,\sqrt{2},1\right]{\left({\left[0,1,-1,0\right]}^{T}\right)}^{2}\\ \; & =\frac{1}{4}\left[1,0,\sqrt{2},1\right]{\left[0,1,1,0\right]}^{T}\\ \; & =\frac{\sqrt{2}}{4}\approx 0.35. \end{array}$$

Now, suppose $y={\left[\frac{1}{2},\frac{1}{2},0,0\right]}^{T}$ (representing a prediction at position $\left(0,\frac{1}{2}\right)$) and $\hat{y}={\left[\frac{1}{4},\frac{1}{4},\frac{1}{4},\frac{1}{4}\right]}^{T}$ (representing the ground truth position $\left(\frac{1}{2},\frac{1}{2}\right)$). Clearly, the prediction in this example is better than the previous example. The product $y^{T}W=\left[\frac{1}{2},\frac{1}{2},\frac{\left(1+\sqrt{2}\right)}{2},\frac{\left(1+\sqrt{2}\right)}{2}\right]$, and the loss is
$$\begin{array}{cc} L & =\frac{1}{4}\left[\frac{1}{2},\frac{1}{2},\frac{\left(1+\sqrt{2}\right)}{2},\frac{\left(1+\sqrt{2}\right)}{2}\right]{\left({\left[\frac{1}{4},\frac{1}{4},-\frac{1}{4},-\frac{1}{4}\right]}^{T}\right)}^{2}\\ \; & =\frac{1}{4}\left[\frac{1}{2},\frac{1}{2},\frac{\left(1+\sqrt{2}\right)}{2},\frac{\left(1+\sqrt{2}\right)}{2}\right]{\left[\frac{1}{16},\frac{1}{16},\frac{1}{16},\frac{1}{16}\right]}^{T}\\ \; & =\frac{2+\sqrt{2}}{64}\approx 0.05. \end{array}$$

As expected, the loss in the second example is less than that of the first example.

### 2.7. Experiments

Eight experiments were conducted between 17 and 20 February 2021, on the Keweenaw Waterway near Michigan Technological University. Figure 6 shows the experimental setup. The Keweenaw Waterway is a narrow and shallow channel of water (a canal), which causes many multipath reflections and scattering. The ice was between 0.4 and 0.5 m thick, and the water was between 6 and 8 m deep. The first three experiments (one on February 17 and two on 18) had snow above the ice, insulating the ice, which caused an uneven, thin layer of slush. By February 19th, high winds had removed the snow, and the surface ice hardened again, so the remaining five experiments were conducted in a hard ice environment.

A snowmobile drove back and forth in front of our sensors to represent a moving acoustic source. A handheld GPS on the snowmobile kept track of the position of the snowmobiles. The two AVSs passively recorded the noise from the snowmobile, which included engine intake and exhaust, as well as track–ice structural–acoustic interaction, for the purpose of localization.

After the data were synchronized, trimmed, and labeled, a total of roughly 3.2 h—11,526 s—of snowmobile acoustic data were recorded on the two AVSs. The position of these AVSs were kept constant, 30 m apart, on either end of the dock next to the Great Lakes Research Center.

#### 2.7.1. Data Explanation

The acoustic data were recorded in time-series at a sample rate of 17,067 Hz using a National Instruments cRIO-9035 with NI-9234 data acquisition cards. The sample rate was set to 17,067 Hz since the sensor’s 3 dB cutoff frequency was at 8000 Hz; thus, frequencies above 8000 Hz were not used in post-processing. The data were transformed into an azigram using Equations (Equation 1)–(Equation 3). The STFT used a Hanning window, 50% overlap, and a segment size of 1706 samples to yield a time step of 0.05 s. Figure 7 shows an example of the azigram of the first 100 s of data. Note there are two snowmobile passes in the azigram, around the 40 and 85 s marks. The snowmobile drives by the sensor around the 40 s mark (heading eastwardly), turns around, then drives by the sensor again near the 85-second mark (heading eastwardly again).

The truth data, being GPS data, were recorded at 1 Hz using a handheld GPS receiver. Figure 8 shows the GPS data through all the experiments. The GPS data were then linearly interpolated, resulting in an upsampling of 20 times, to match the sample rate of the azigram data.

To prepare the data for input to the neural network, the azigram was linearly normalized from its [−π,π] range to [0,1] and the GPS data were linearly normalized with the total maximum and minimum latitude and longitudes set to the interval [0,1]: latitude was normalized from $\left[47.1200^{\circ },47.1225^{\circ }\right]$ to [0,1] and longitude from $\left[-88.548^{\circ },-88.542^{\circ }\right]$ to [0,1]. For classification networks, the GPS data were processed with Equation (Equation 16) with k=100 to represent a 10×10 grid of latitude and longitude position.

### 2.8. Network Explanations

The neural networks process the same data, i.e., the data are pre-processed in the same manner for every neural network. The azigram data are shaped to use the prior 512 time steps for a single prediction. Each AVS’s azigram frequencies are downsampled to contain 256 frequency bins. The two AVS’s frequency data are concatenated along the frequencies; hence, the input data are a 512×512 azigram in the neural network. Each network predicts a single output value, y, at the final time step of the 512×512 sample. In other words, the networks’ input data are a sliding window of 512 samples, and each network predicts the new location at the end of the 512 window, then the window is moved forward by 1 sample from a time window of [n,n+512] to [n+1,n+513].

We compared four large neural networks and four small neural networks. The small neural networks are demonstrated as a simpler method in localizing an acoustic source; less training time, less training data collection, and less calculation time are required for “small” networks. Because our dataset is very large, we also explored large neural networks, though this may not be practical for situations where data collection is difficult or impossible to achieve due to budget limitations, lack of available data, or time limitations in labeling, or the environment is not complex enough to require such a large network.

The four large networks are the following: a ResNet50 [24] CNN-based network, an LSTM-based network, a Transformer-based network [15], and a ViT-based network [13]. The four small networks have an arbitrary requirement to contain less than 1 million parameters to give a fair comparison. Figure 9 and Figure 10 show a comparison of each of the networks. The difference between a “small” and “large” network is adjusted in the architectures by the ×N value in both Figure 9 and Figure 10, i.e., *N* is smaller in small networks. Table 1 shows the number of layers *N* for each of the neural networks, and Table 2 contains the number of trainable parameters for each neural network.

The categorical classification neural networks predict a probability of each grid location class. This classification network predicts its results in a softmax activation function—Equation (Equation 13)—to assert a probability output. The benefit of the categorical classification neural network is its opportunity to add uncertainty to its prediction.

The ordinal classification neural network predicts exactly the same type of output as the categorical classification network, but rather than using the mean-squared error loss function in Equation (Equation 19), the network uses the ordinal loss function in Equation (Equation 20).

### 2.9. Training and Hyperparameters

Each network used the Adam optimizer [28] with a learning rate of 0.0001 and parameters $\beta_{1}=0.9$ and $\beta_{2}=0.999$. We batched 32 samples of size 512×512 in a single backwards propagation step. Each batch had its data randomized except for the LSTM, where batches were sequential to support the LSTM’s long-term memory. Seven of the 8 experiments were used as training data, consisting of a total of 189.868 samples, i.e., roughly 2.6 h of data. Ten percent of the training data were used to validate the model, i.e., 18.987 samples. The model weights with the lowest loss using this validation set were then tested on the test data, which we can now show.

## 3. Results

The data on which we tested our algorithms consisted of an experiment where a single snowmobile moved by the sensors back and forth on February 17. There were 39.628 samples, i.e., 1.981 s, and no neural network was trained on any data from this day to isolate the training and test data.

The neural networks were programmed in Python using the Tensorflow backend and Keras frontend to create these models [29,30]. The networks were trained using an NVIDIA GeForce RTX 3090.

The accuracies of each neural network and their respective loss functions are shown in Table 3. Notice the ViT has almost over a 10-fold increase in accuracy. When comparing the two sizes of networks, the training times for each are telling, tabulated in Table 4. The timing differences between each model are significantly different, except for the large and small CNN models.

The results may be misunderstood simply reading Table 3 and Table 4. For a visual representation of our data, we will start off with the predicted coordinates for what the MSE actually represents. Figure 11 shows a section of a time-series representation of the data using both the latitude and longitude positions of the snowmobile and each algorithms’ predicted positions of the snowmobile. Figure 11 can be misleading where one may see the CNN and Transformer networks seem to be on-par with, or close to, the results of the ViT. Mapping these results to an −x,−y plane, Figure 12 shows a more critical view for what these small amounts of errors indicate. Even with a top-down view of the experiment, the ViT tracks the snowmobile at high accuracy in comparison to the other models. For the ViT, it should be noted that its mean accuracy is 2.9 m. This is very close to the accuracy of our GPS receiver: the reported 95th-percentile mean error is 2.545 m in Minneapolis, Minnesota, which is relatively near Houghton, Michigan, from 1 January to 31 March 2021 [26], and the GPS was recorded in a relatively open area. Therefore, the ViT appears to have reached the maximum achievable accuracy of our experimental truth data. That is, our truth data are not accurate enough to verify errors significantly better than 2.9 m. These significant results are further discussed in Section 4. Almost all of the test data are similar to Figure 11 and Figure 12, shown in Appendix A Figure A1.

Although most test data are similar, there exists a section of the test data where a snowmobile idles (does not move) for 25 s, and the networks perform relatively poorly with these data. Figure 13 shows the predicted locations from each network at the time where the snowmobile is idling (not moving) in a bird’s eye view. The Transformer, CNN, and LSTM networks all struggle to notice when the snowmobile is idle. Those three neural networks were not able to notice the stationary source and continued to predict movement. Note that the LSTM seems to follow a circular pattern, which indicates the network is anticipating the snowmobile to drive in this pattern.

## 4. Discussion

The core structure behind the network architectures described in Section 2.2, Section 2.3, Section 2.4 and Section 2.5 is indicative of the results shown in Section 3. Similar to our results, ViTs have shown excellent results in image classification [13]. What may be surprising or not intuitive is the magnitude by which the ViT performance surpassed all other models, most surprisingly the similarly structured Transformer. Each neural network tracks the general trend of the snowmobile position, while the ViT tracks the positions almost perfectly. To explain this, the Transformer determines attention for each input sample individually, and the ViT attends to subsections of the input data. The input data have 512 samples of dimension 512, and the Transformer attends each time step to all the other time steps. This produces a significant amount of attention solely within the Transformer block. On the other hand, the ViT positionally embeds the 512 samples so that attention is made in a temporal and frequency connection. The embeddings also reduce the attention matrix with the size of patches per Transformer network. These reduced attention matrices allow for a “deeper” model with the ViT network.

The original Transformer well describes *Natural Language Processing* (NLP) [15] with its connection between word embeddings, but this does not transfer well to spectrum analysis. The Transformer network does not allow for attention along the time and frequency domain, which is addressed in modified Transformer papers [31]. Specifically, the ViT is a type of modified Transformer, which, in our example, embeds the 512 attention parameters into 64 smaller regions of interest. These areas are trainable, and the embedded positions allow the ViT to attend to time and frequency patterns rather than solely time. Additionally, the embedded positions yield a smaller number of positions to which the ViT attends. This embedding helps scale the ViT to attend to higher-dimensional data with the lower amount of attention values. Even with this explanation, the significant increase in accuracy exhibited by the ViT is remarkable. Similar behavior in results is summarized in surveys of ViTs [17,32]; notably, the results in [33] show major improvements in image classification accuracy using the ViT.

The ordinal classification approach for localization is not a panacea for every localization problem. The ordinal classification approach shows improved results in our experiments and gives way for soft-labeled truth data when the truth data are not absolutely accurate. The network is capable of predicting with low confidence, although our training data do not facilitate the networks utilizing this yet. An important question arises: Why are some networks better than others with different loss functions? We believe that the LSTM and Transformer networks are most suited for regression, as LSTMs were constructed for time-series data [11] and Transformers were developed for NLP [15]. The opposite is true for the CNNs and ViTs. The CNN and ViT architectures are suited for classifying images; hence, it is understandable why the CNN and ViT performs better for the classification approaches we propose for localization.

The results for the large neural networks are impressive, but it may be unacceptable or impractical to use such large networks in real-world scenarios. For example, in an remote embedded system, using an 85 million parameter network such as the large ViT would be wasteful with power consumption to calculate all the operations. Additionally, the number of data points required to train such large neural networks costs an extraordinary amount of time and effort to produce, as well as using an expensive GPU to train the network. In contrast, the small neural network results are a more practical view for the number of parameters to be used in a real-world scenario. Therefore, it is important to look at the loss in accuracy as a friendlier, real-world use case with smaller networks.

## 5. Conclusions

In this paper, we developed different neural networks for processing data from a pair of underwater AVSs. The sensors recorded a moving anthropogenic acoustic source, and the data were analyzed using different neural networks to estimate the location of the target. Each network—the CNN, LSTM, Transformer, and ViT—all tracked the position relatively well, but when comparing the networks, we found that the ViT predicted source location with excellent accuracy, an order of magnitude more accuracy. The ViT was able to analyze our highly dimensional data and track the acoustic source well. Additionally, the networks were reduced to have a smaller number of parameters in order to compare the loss in accuracy.

Finally, we studied three approaches to localizing a moving target. A regression loss function was the baseline method to compare with our non-conventional methods: a categorical classification and ordinal classification approach for localization. We showed that the ordinal classification approaches performed better for networks better suited for classification, being the CNN and ViT. The regression loss function performed better for the networks better suited for time-series data, being the LSTM and Transformer.

## Figures and Tables

**Figure 1 sensors-22-04703-f001:**
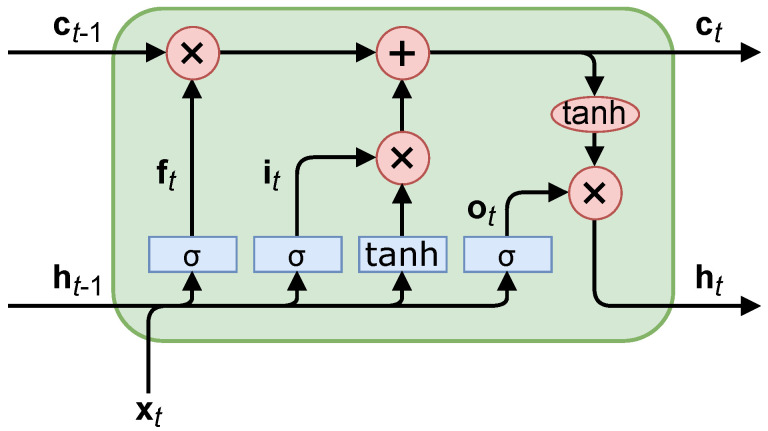
A long short-term memory cell, where blue rectangles indicate trainable parameters and red ovals indicate a math operation (non-trainable).

**Figure 2 sensors-22-04703-f002:**
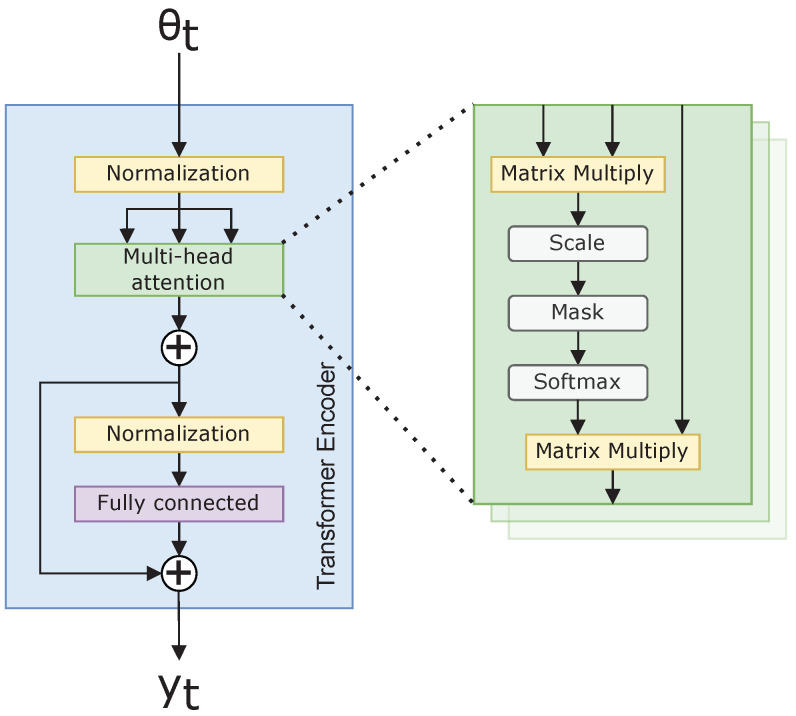
Transformer neural network encoder.

**Figure 3 sensors-22-04703-f003:**
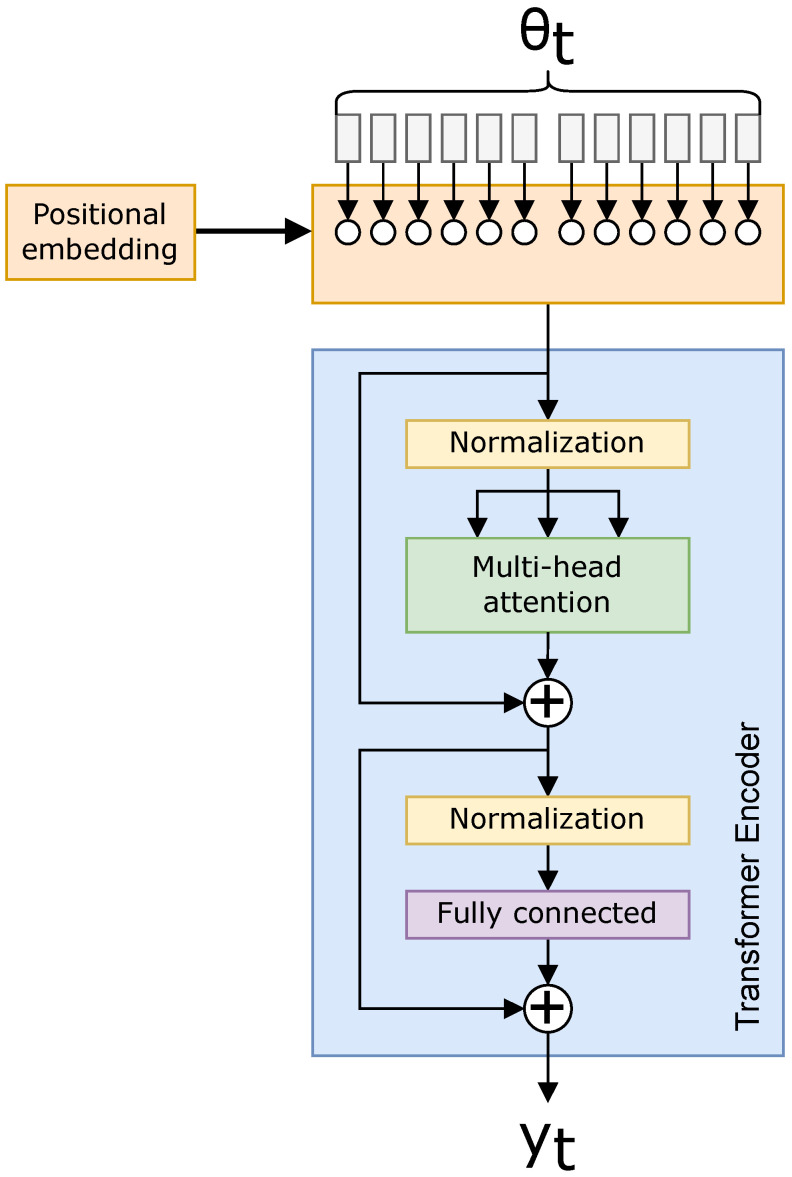
Example *Vision Transformer* (ViT) where our input data are positionally embedded prior to passing into the Transformer, being Equations (Equation 12)–(Equation 14).

**Figure 4 sensors-22-04703-f004:**
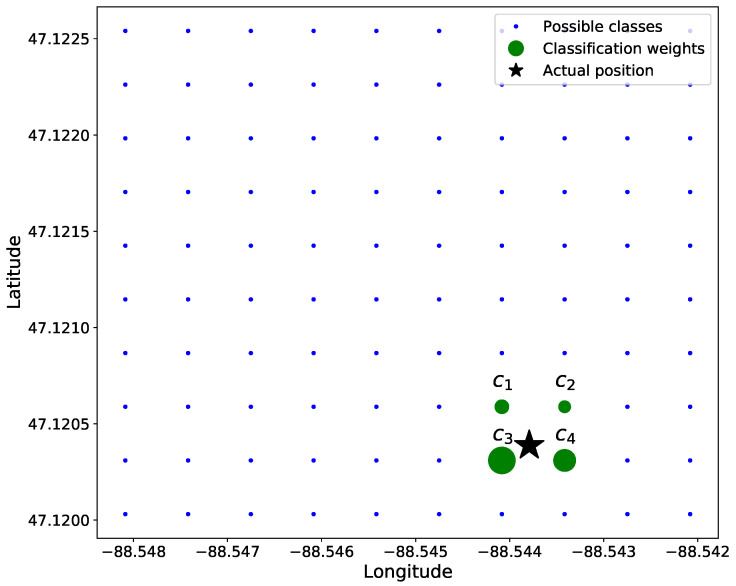
Soft classification of linear position where Δ=1. The star is the original position, and the circle size corresponds to the weight of each value.

**Figure 5 sensors-22-04703-f005:**
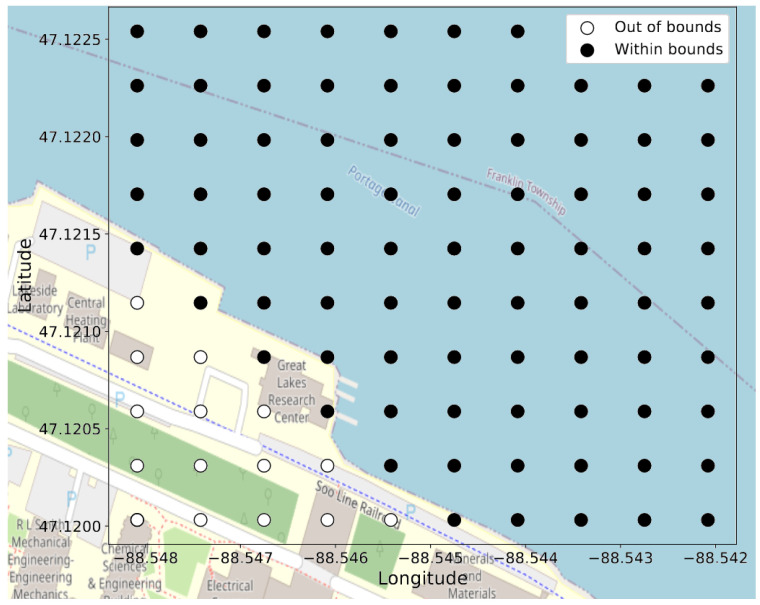
Eighty-five classes, out of the possible 100, for a 10×10 grid where the training and test data are not present in any of the “out of bounds” labels.

**Figure 6 sensors-22-04703-f006:**
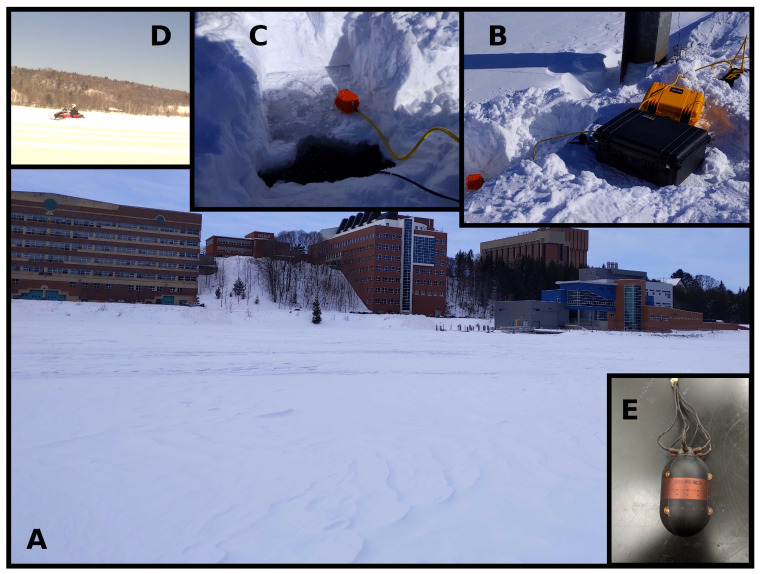
Conditions under which the experiments were conducted: (**A**) shows the Keweenaw Waterway frozen over, looking SSW at Michigan Technological University; (**B**) shows the sensors and data acquisition system on a dock near the Great Lakes Research Center; (**C**) shows a close-up of where the sensors are deployed in the water; (**D**) shows a snowmobile driving in one of the experiments; (**E**) is a close up of the AVS.

**Figure 7 sensors-22-04703-f007:**
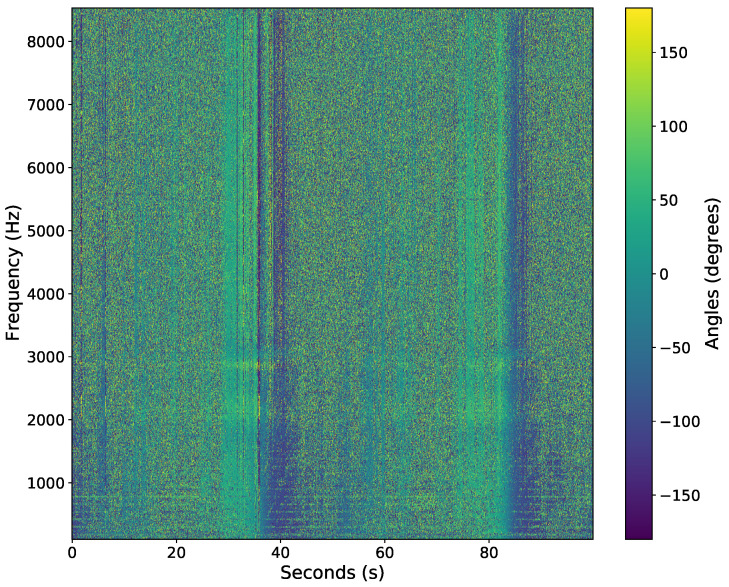
Azigram response from a single AVS of a snowmobile driving past the AVS at roughly 40 and 85 s.

**Figure 8 sensors-22-04703-f008:**
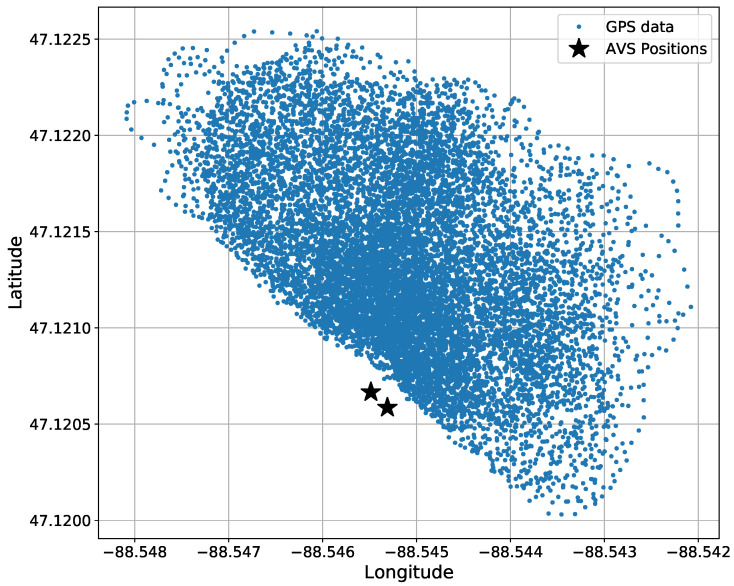
Bird’s eye view of the total amount of GPS data in all datasets before 20-times interpolated. GPS data are accumulated from 8 experiments. The two AVS positions are shown for a reference.

**Figure 9 sensors-22-04703-f009:**
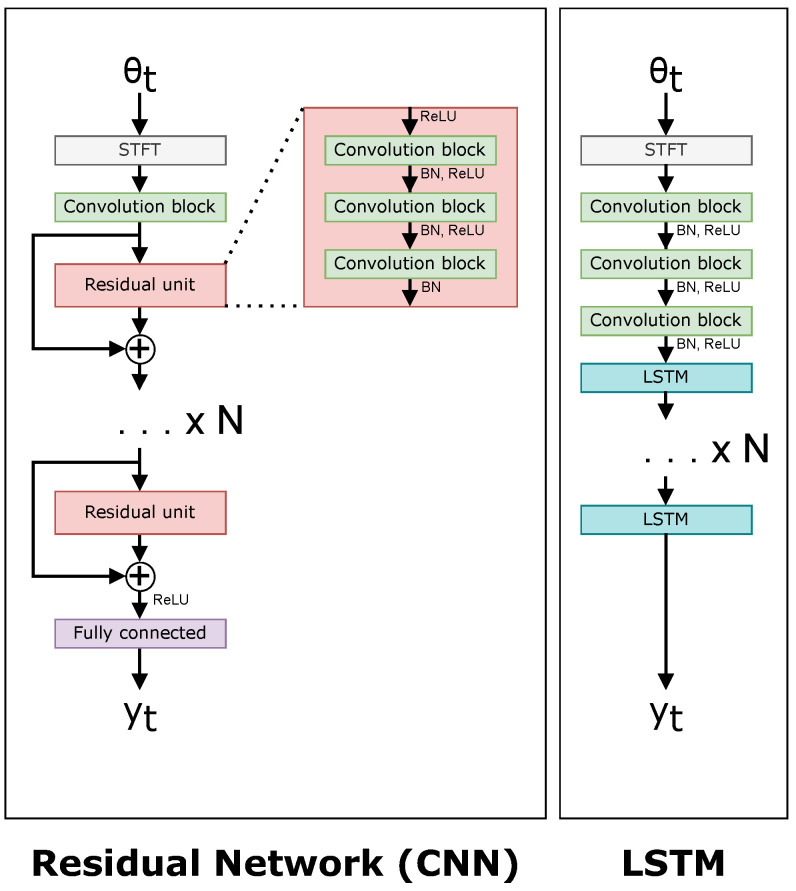
Network architectures for the CNN (**left**) and LSTM (**right**).

**Figure 10 sensors-22-04703-f010:**
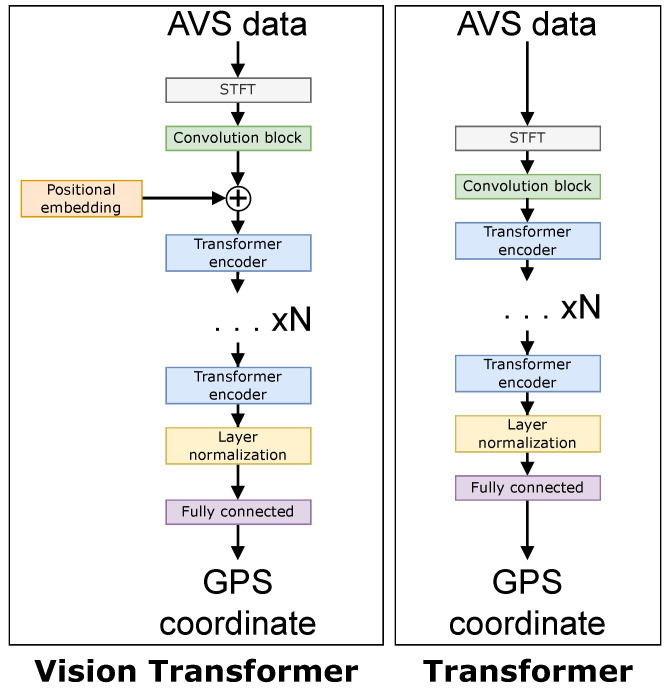
Network architectures for the ViT (**left**) and Transformer (**right**).

**Figure 11 sensors-22-04703-f011:**
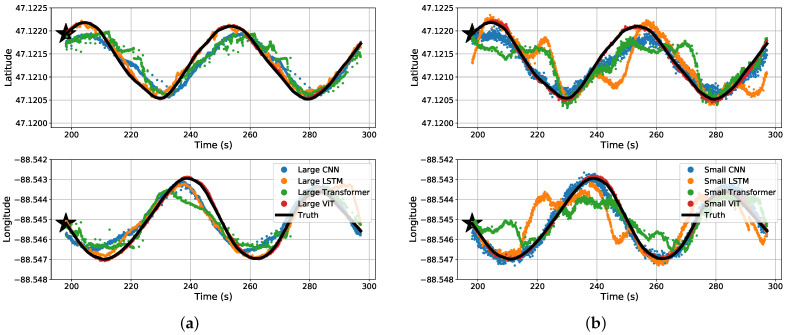
Time-series split predicted results for the four different (**a**) large regression algorithms and (**b**) small regression algorithms.

**Figure 12 sensors-22-04703-f012:**
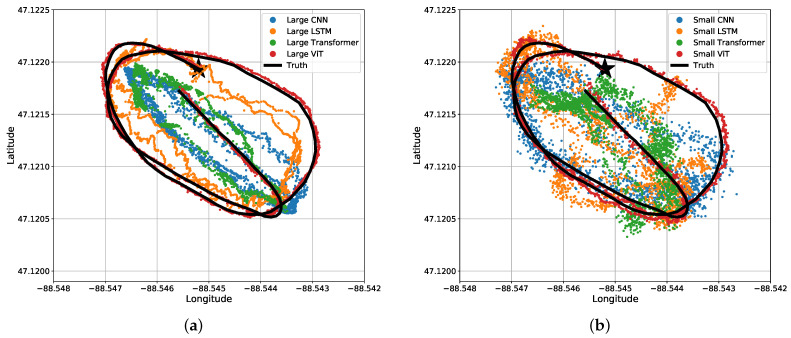
Bird’s eye view of results for the four different (**a**) large regression algorithms and (**b**) small regression algorithms. The same data and predictions from Figure 11 are shown.

**Figure 13 sensors-22-04703-f013:**
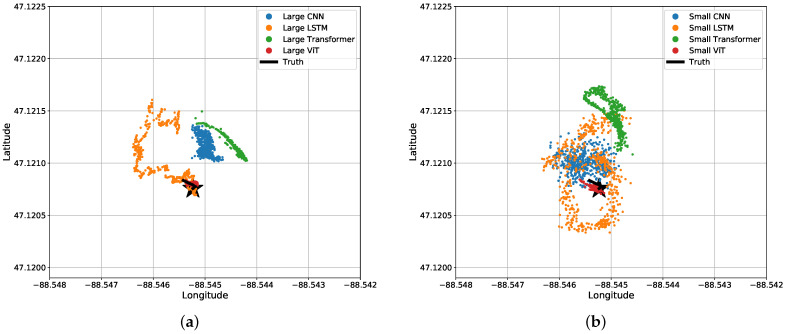
Bird’s eye view of results for the four different (**a**) large regression algorithms and (**b**) small regression algorithms when the acoustic source is stationary for 25 s.

**Table 1 sensors-22-04703-t001:** Depths of the backbone for each type of network shown in Figure 9 and Figure 10.

	CNN	LSTM	Transformer	ViT
Large	16	5	8	12
Small	4	1	1	8

**Table 2 sensors-22-04703-t002:** Total trainable parameters for each neural network architecture.

	CNN	LSTM	Transformer	ViT
Large	23,849 k	13,825 k	16,911 k	85,846 k
Small	892 k	905 k	843 k	928 k

**Table 3 sensors-22-04703-t003:** Neural network results on test data from February 17. The results indicate the mean distance in meters between the predicted results by the neural network and the recorded results by the GPS. A ±1σ deviation is shown.

	CNN		LSTM	
	Large	Small	Large	Small
Regression	39.3± 29.1	27.1± 21.7	44.2± 53.9	58.7± 62.7
Categorical	26.7± 57.3	28.4± 27.0	49.4± 47.8	41.9± 40.0
Ordinal	21.4± 31.2	33.1± 35.2	64.6± 49.2	68.2± 54.3
	**Transformer**		**ViT**	
	**Large**	**Small**	**Large**	**Small**
Regression	42.1± 30.3	65.7± 48.8	4.9± 3.7	5.9± 4.8
Categorical	49.8± 39.3	44.5± 45.1	3.1± 2.5	3.7± 3.0
Ordinal	53.9± 45.0	44.5± 45.1	2.9± 2.5	6.7± 6.1

**Table 4 sensors-22-04703-t004:** Neural network mean training times per epoch.

	CNN		LSTM	
	Large	Small	Large	Small
Regression	671 s	654 s	2358 s	1931 s
Categorical	620 s	657 s	2170 s	1933 s
Ordinal	675 s	656 s	2150 s	1930 s
	**Transformer**		**ViT**	
	**Large**	**Small**	**Large**	**Small**
Regression	1358 s	639 s	1700 s	654 s
Categorical	1188 s	648 s	1785 s	658 s
Ordinal	1070 s	647 s	1752 s	660 s

## Data Availability

The data used in this research were recorded by the Meggitt *VS-209*, which is under International Traffic in Arms Regulations (ITAR) and cannot be released to the public.

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
