# Peer review of "Through-Ice Acoustic Source Tracking Using Vision Transformers with Ordinal Classification"

_sensors, 2022, doi:10.3390/s22134703_

Round 1

Reviewer 1 Report

This article is complete and meets the requirements of the Sensors journal and the language of the paper overall is good. So, the manuscript may be considered for publication in Sensors journal in its present form.

Author Response

Other reviewers have requested fixes in grammar and reducing the size of the results section, which have been both done.

Reviewer 2 Report

1. The authors compare different deep learning networks, such as Transformers, Convolutional Neural Networks (CNNs), Long Short-Term Memory (LSTM) networks, and Vision Transformers (ViT), for passive localization and tracking of single moving, on-ice acoustic sources using two underwater acoustic vector sensors.

2.     Please demonstrate the innovative concepts of manuscript in detail.

3.   The manuscript has 21 figures; the number of the figures should be decreased.

4.     Revise the English thoroughly before submission.

Author Response

2. Done, thank you.
3. Another reviewer asked the same but with more specifics: there are now 14 figures, 3 are in the results, 1 in the appendix.
4. Done, thank you.

Reviewer 3 Report

This is an interesting study that compared different neural networks for processing data from a pair of underwater acoustic sensors which were used to record moving anthropogenic acoustic sources. The ViT was demonstrated to perform the best with the highest accuracy. After a thorough review of this manuscript, I would like to request the authors make a major revision to the manuscript, because this paper suffers from several flaws. Below are my specific comments on the manuscript:

  1. The authors should clarify how their methods deal with the complex sound speed profile in water. 
  2. The authors mentioned that "a long short-term memory (LSTM) neural network have been shown [9–12]" but didn't provide any comments on its strength or limitation. 
  3. The authors first introduced the "ViT" in line 31, which is the most important concept in this study. However, the references for this concept [13-16] are online pre-print and are no longer available when I tried to access them. Without these most important references related to the key method used in this study, I cannot assess the scientific soundness and the reliability of the results. 
  4. In line 33, the authors mentioned that "To date, there has been no paper published on ViT-based localization for through-ice acoustics or underwater acoustics", so what was Vit famous for and what's the original application of Vit?
  5. In line 35, the authors first introduced classification and regression, but without any detailed background. 
  6. In line 36, "This models the problem as a classification problem within a grid of positions to localize the source", what is the "problem" here? The authors haven't defined what the problem is. 
  7.  In line 52, the authors said their acoustic sensor is a "new type of sensor". However, the 2 references the authors cited here [17][18] were published in 1995 and 2004, respectively. I didn't see any reason the authors should claim the sensor as a "new type".
  8.  The authors should calibrate the sensor and characterize the bandwidth and sensitivity before any measurement. Without any calibration, the data is not reliable.
  9. In line 59-61, the description lacks reference on the typical bandwidth of snowmobile and the acoustic sensor. 
  10. Check the grammar in this sentence "This angle measurement tells us from which direction the sound arrives, be it from the acoustic source we are trying to track, or be it any other source". 
  11. Reference is needed in 2.5. and 2.6.1.
  12. In line 182, "For the purposes of this paper, the errors in the truth data and its distribution are not considered because the uncertainties of our truth data are smaller than the distances between each class", the authors should show raw data and elaborate why the uncertainty of the truth data is smaller than the distance between each class. 
  13. In line 194, "but for this study we simply constrained the classification grid to within the banks of the region of study (a canal).", what's the justification of this constrain? Will this cause any loss of the data? What's the limit?
  14. The results and figures are very massive. The authors should concentrate their key results and the most important information. 

I hope my comments could be helpful.

Author Response

1. Yes, this has been clarified more, in addition to all of the other complexities in underwater acoustics that were assumed, but never directly explained.
2. The paper's been edited to has the LSTM's negatives and benefits introduced in the introduction before being explained more fully in the rest of the paper.
3. Thanks, the references are fixed. There were other references outside of those four that were also preprint, so those are fixed now too, and we added DOI links to every reference that has one.
4. ViTs were used for computer vision and a survey on different approaches is added, with a specific one used in CV.
5. Thanks, this should be more clear hopefully. Please let us know if it isn't.
6. Thanks for the clarity check. It should be the continued localization problem described prior. Hopefully this edit will make it more clear.
7. You're right on that, yes. We removed this phrase to prevent confusion.
8. The Meggitt VS-209 is calibrated in a lab before purchase and never removed from its casing (which would also destroy it, as well as requiring recalibration). The sensitivity and exact bandwidth are sadly ITAR and cannot be discussed, I wish I could show them, but I have written that the upper 3-dB drop off occurs at 8 KHz.
9. Citation added for snowmobile bandwidth and explanation added for this.
10. Rewritten second part of phrase to be less confusing.
11. Citations are added.
12. We're not sure why raw data is required, but the experiment does not have more than simple GPS coordinates from a handheld GPS. Figure 8 with the GPS data technically shows the raw data, but there's no analysis on how accurate our GPS data is besides looking at the accuracies noted by NOAA for our testing period. We've added that the accuracies by NOAA are significantly lower than the distances between each classes, but we're sorry, we don't understand why raw data is required.
13. There's more explanation added in the paper to each of your comments, but to answer your questions directly: a boat is unable to leave the water and drive up on land, so there's no point to add this excess dimensionality to our localization problem. The same is true for our snowmobile experiment because the ice near the edge of the canal was very thin and the snowmobile would break through the ice, so they stayed away from the edges, but that is too specific to our exact environment.

No data was removed, only "impossible" classes were removed, being those on the land. It could cause data loss, but that would be an overly constrained example, the time-series data would be spliced, and the neural network would have to handle jumping locations instantly.

For your last question, we don't exactly know what you mean, but one could engineer the problem by removing all classes that aren't exactly with the data. That is a very unfair and biased method to analyze the results, so we did not do this. One could do this in a real world environment when using as a product (such as tracking locations only within a shipping lane). We find it's better to be objective in the results, even if one could possibly get better results by constraining even more.

14. Appendices are changed to have 1 plot with multiple subfigures. We find this figure important to show we are not cherry-picking our results. We also removed a plot in the results that was not entirely necessary and not discussed much. Thank you.

Round 2

Reviewer 2 Report

no further comment.

Reviewer 3 Report

The authors have addressed my comments and provided the necessary information, so I would recommend acceptance for publication.